# Neuroprotective Strategies in Aneurysmal Subarachnoid Hemorrhage (aSAH)

**DOI:** 10.3390/ijms22115442

**Published:** 2021-05-21

**Authors:** Judith Weiland, Alexandra Beez, Thomas Westermaier, Ekkehard Kunze, Anna-Leena Sirén, Nadine Lilla

**Affiliations:** 1Department of Neurosurgery, University Hospital Würzburg, Josef-Schneider Str. 11, 97080 Würzburg, Germany; Beez_A@ukw.de (A.B.); Westermaie_T@ukw.de (T.W.); Kunze_E@ukw.de (E.K.); Siren_A@ukw.de (A.-L.S.); 2Department of Neurosurgery, Helios-Amper Klinikum Dachau, Krankenhausstr. 15, 85221 Dachau, Germany; 3Department of Neurosurgery, University Hospital Magdeburg, Leipziger Str. 44, 39120 Magdeburg, Germany

**Keywords:** subarachnoid hemorrhage (SAH), inflammation, thromboinflammation, metabolism, neuroprotection, therapy

## Abstract

Aneurysmal subarachnoid hemorrhage (aSAH) remains a disease with high mortality and morbidity. Since treating vasospasm has not inevitably led to an improvement in outcome, the actual emphasis is on finding neuroprotective therapies in the early phase following aSAH to prevent secondary brain injury in the later phase of disease. Within the early phase, neuroinflammation, thromboinflammation, disturbances in brain metabolism and early neuroprotective therapies directed against delayed cerebral ischemia (DCI) came into focus. Herein, the role of neuroinflammation, thromboinflammation and metabolism in aSAH is depicted. Potential neuroprotective strategies regarding neuroinflammation target microglia activation, metalloproteases, autophagy and the pathway via Toll-like receptor 4 (TLR4), high mobility group box 1 (HMGB1), NF-κB and finally the release of cytokines like TNFα or IL-1. Following the link to thromboinflammation, potential neuroprotective therapies try to target microthrombus formation, platelets and platelet receptors as well as clot clearance and immune cell infiltration. Potential neuroprotective strategies regarding metabolism try to re-balance the mismatch of energy need and supply following aSAH, for example, in restoring fuel to the TCA cycle or bypassing distinct energy pathways. Overall, this review addresses current neuroprotective strategies in aSAH, hopefully leading to future translational therapy options to prevent secondary brain injury.

## 1. Introduction

Aneurysmal subarachnoid hemorrhage (aSAH) is a complex cerebrovascular disease with profound systemic complications, accounting for about 5% of all strokes. The worldwide incidence of aSAH is approximately 700,000 person–years with an overall mortality of about 40% [1,2]. Brain injury following aSAH is multimodal and occurs directly, as early brain injury, however, also secondarily, as delayed brain injury [3]. Angiographic cerebral vasospasm (CVS) occurs in approximately 70% of patients during the first 2 weeks after aSAH, but the incidence of delayed cerebral ischemia (DCI) is only around 30%, with DCI remaining the major cause of morbidity and mortality among patients who survive the initial treatment of the ruptured aneurysm [4,5]. Several mechanisms during the acute phase of SAH contribute to DCI and poor outcome. These include neuroinflammation, microthrombosis, cortical spreading depolarizations, disrupted integrity of the blood–brain barrier (BBB), microvascular dysfunction and metabolic derangement [6,7,8,9].

Despite the extensive advances in experimental research, especially in recent years, translational clinical therapy and data are still lacking, persistently motivating researchers all over the world to find new strategies for neuroprotective therapy following aSAH. The main goal of this review was to summarize the role of neuroinflammation, thromboinflammation and metabolism following aSAH with a special focus on new neuroprotective targets in these fields, possibly leading to translational new approaches in clinical neuroprotective therapy.

## 2. Role of Neuroinflammation in aSAH

### 2.1. Activation of the Immune System

The initial global hypoperfusion after aSAH leads to inflammatory processes, which occur in blood vessels as well as in cerebrospinal fluid (CSF). While talking about inflammation in a non-infected environment, persistent actions of the innate immune system were predominantly found [10,11,12].

Peripheral monocytes invade the brain as macrophages. Lymphocytes, as part of the innate and adapted immunity, are highly activated in contrast to B- and T-cells as products of the adapted immune system, which are just rarely upregulated [13]. Since macrophages and neutrophils enter the subarachnoid space, they degranulate, whereby inflammatory factors are released [14].

An early gain of neutrophils, already 3 days after aSAH, was found to be associated with a later vasospasm. Therefore, it can be concluded, that the mechanism of neutrophil signaling ameliorates vasospasm.

Inflammatory cells and associated cytokines as well as their receptors are upregulated in the subarachnoid space, while entering from within the blood vessels and acting on vascular walls formed by the increased reactivity of microvessels to post-hemorrhagic CSF [15].

An increased permeability of the BBB enables cytokines to reach the brain parenchyma and also circulating immune cells to access the perivascular spaces and reach the brain interstitium [16,17]. Additionally, an increased cytokine production as a response to injured brain parenchyma and the increased permeability of the BBB leads to global cerebral edema.

### 2.2. Acute Events Following aSAH

After the initial aneurysm rupture, blood leaks into the subarachnoid space. The breakdown of red blood cells and degradation over time leads to the deposition of hemoglobin. As a result of red blood cell breakdown, methemoglobin, heme and hemin can lead to activation of Toll-like receptor 4 (TLR4), which signals inflammatory cascades that damage neurons and white matter [18]. Due to the neuroimmunological damage of neurons, a possible link with metabolic derangement may exist. Via the release of redox-active iron, Hemin has been linked to alter the balance of oxidants and anti-oxidants. The redox-active ion produces superoxide and hydroxyl radicals as well as lipid peroxidation while depleting anti-oxidant stores such as nicotinamide adenine dinucleotide phosphate (NADPH) and glutathione [18,19].

In addition, immunomodulatory cells, notably microglia, are activated due to leaking blood after aneurysm rupture. These cells trigger the upregulation of numerous cell adhesion molecules within endothelial cells, which subsequently allows a multitude of inflammatory cells to bind and enter the subarachnoid space [18,20]. Once these inflammatory cells, such as macrophages and neutrophils, enter the subarachnoid space, they phagocytize the extravasated, degrading blood cells in an effort to clear free hemoglobin and therefore promote neurostability and recovery. By the binding of hemoglobin to haptoglobin, a rapid engulfment by immune cells is facilitated [18].

### 2.3. Subacute/Chronic Events Following aSAH

As described above, peripheral immune cells such as neutrophils and macrophages are attracted to clear free hemoglobin after aneurysm rupture. Moreover, they might also become trapped in the subarachnoid space due to alterations in CSF flow and the restoration of the endothelial tight junction barrier. During degranulation in the subarachnoid space, macrophages and neutrophils release a multitude of inflammatory factors, including endothelins and oxidative radicals. Further downstream, these factors can cause inflammation-induced vasoconstriction, meningitis and cerebritis [21]. It remains unclear whether neutrophils and macrophages are passing through an intact or disrupted BBB while being recruited.

### 2.4. Microglia

The microglia can be understood as the on-site phagocytes of the central nervous system (CNS), which are able to provoke an upregulation of inflammatory cytokines, especially of interleukins IL-1ß and IL-6 as well as of tumor necrosis factor alpha (TNFα) as a response to infection or even hemorrhage due to an accumulation near the vessel rupture with release into both serum and CSF [22,23,24].

This increased production causes the destabilization of the blood–brain barrier by a wave of intraparenchymal inflammation, which causes neuronal injury where it has typical pro-inflammatory cytokines, even if aSAH is primary a bleeding outside the brain parenchyma. The breakdown of red blood cell products and its release of inflammatory cytokeratins could be understood as a trigger of vasospasm and tissue injury [14].

During the early phase after aSAH, as already mentioned above, the breakdown and degradation of red blood cells leads to a hemoglobin detachment. Subsequently, methemoglobin, heme and hemin facilitate a neuroinflammatory reaction as they are able to activate TLR4, which are expressed in microglia [25,26]. TLR4 generates among others an inflammatory response by interacting with the nuclear factor kappa-light-chain-enhancer of activated B cells (NF-kB), which results in damaging nerve tissue [27]. In summary, different pathways linked to TLR4 activation were found. Therefore, new strategies for immunotherapy that target microglia and TLR4 signaling should be investigated [28].

Microglia accumulation plays a significant role in the cerebral spreading of inflammation [29] as it can be a significant target for treatment strategies. The amount of neuronal cell death can be reduced by microglia depletion. The intracerebral accumulation of inflammatory cells occurs between day 4 and 28 after aSAH, contributing to delayed brain injury [30].

### 2.5. Endocannabinoids

Endocannabinoids (eCBs) are lipid-soluble molecules and the endogenous analogs of the psychoactive constituents of cannabis plants. They are produced on demand in response to increased intracellular calcium levels. Arachidonoylethanolamine, also known as anandamide (AEA), and 2-arachidonoyl-glycerol (2-AG) are the main representatives of eCBs. The endocannabinoid system (ECS) is described as a complex lipid network consisting of cannabinoid receptors (CBRs), endogenous ligands and the enzymes involved in endocannabinoid degradation and synthesis [31,32]. The ECS has been implicated as an important regulatory component by the regulation of the local CBF and by the maintenance of the BBB and related transport proteins [32].

All major cell types involved in cerebrovascular control pathways (i.e., smooth muscle, endothelium, neurons, astrocytes, pericytes, microglia and leukocytes) are capable of synthesizing eCBs and/or the expression of their target proteins like the cannabinoid type 1 receptor (CB1R) and cannabinoid type 2 receptor (CB2R).

This leads to the hypothesis, that the ECS may importantly modulate the regulation of cerebral circulation in physiological and pathophysiological conditions. Experimental data suggest that the direct effect of cannabinoids on cerebral vessels is vasodilatation, at least partially mediated by CB1R. It was shown that, for certain cerebrovascular pathologies like SAH and traumatic as well as ischemic brain injury, the activation of CB2R (and to date unidentified non-CB1R/non-CB2R) receptors may improve brain blood perfusion by attenuating vascular inflammation [32].

Microglia in the homeostatic “resting” state synthesize 2-AG and AEA and express the cannabinoid receptors CB1R and CB2R at low levels. When activated, microglia significantly increase their synthesis of eCBs and upregulate their expression of CB2R, making them produce more neuroprotective and less pro-inflammatory factors [33].

Both synthetic cannabinoids and eCBs were shown to inhibit TNFα release and the release of other cytokines like IL-1α, IL-1β and IL-6 [32].

### 2.6. Metalloproteinases

Metalloproteinases (MMPs) are a family of proteases consisting of multiple subtypes. MMP-9 is the most widely investigated metalloproteinase and has been shown to be responsible for the degradation of tight junction proteins, which are substantial in the maintenance of BBB integrity. Clinical studies of aSAH have reported an elevation of MMP-9 levels in brain tissue, serum and CSF as well as in the vessel wall [34]. Therefore, MMP-9 might be a possible target for neuroprotective therapies by restoring BBB integrity or even prevent BBB disruption in the first place.

### 2.7. High Mobility Group Box 1

The high mobility group box 1 (HMGB1) protein is a pro-inflammatory-like cytokine with an initiator role in neuroinflammation, that has been implicated in traumatic brain injury (TBI) as well as in early brain injury (EBI) after aSAH. An extracellular overexpression of HMGB1 following aSAH has been described. Once reaching the extracellular milieu, HMGB1 serves as a damage-associated molecular pattern (DAMP) protein and exerts an inflammatory response, engaging several inflammatory mediators. After being released from neurons and astrocytes, HMGB1 begins the production of several inflammatory markers, including TNFα, IL-6 and IL-1β [35]. Extracellular HMGB1 interacts with TLR4 and the receptor for advanced glycation end products (RAGE) to initiate cell migration and the production of cytokines [35,36].

HMB1 acts as an inflammatory mediator due to the activation of neuroinflammatory cascades and rupture of the BBB [37,38]. The increased HMGB1 leads to an aggravated inflammation and upregulation of inflammatory mediators via TLRs/NF-kB signaling cascades [39]. An early increase following aSAH with its maximum peak after one day and its proinflammatory role was demonstrated in experimental aSAH by the verification of increased TLR4, IL-1ß as well as NF-kB after the administration of recombinant HMGB1 [40]. By activating Janus-Kinase (JAK2)/signal transducer and activator of transcription (STAT3), it contributes to EBI. Therefore, the JAK2/STAT3 inhibitor AG490 decreased the HMGB1 expression as well as its nuclear to cytosol translocation, which leads to ameliorated EBI in experimental aSAH [41].

The experimental administration of a HGMB1 antibody demonstrated an attenuated progression of CVS due to a reduced upregulation of inflammatory molecules such as TNFα, TLR4 and IL-6 [38]. Accordingly, glycyrrhizin (a HGMB1 inhibiting molecule) reduced CVS by the downregulation of proinflammatory cytokines (IL1-ß, IL-6 as well as TNFα) [42].

Even though no clinical data have been provided to date, targeting the HMGB1 pathways may be a promising therapeutic approach [39]. Elevated serum HMGB1 levels in aSAH-patients from day 1 remained elevated until day 13 in patients developing CVS, reflecting a biomarker potential of HMGB1 [43].

### 2.8. Autophagy and NF-κB-Mediated Inflammation

SAH following aneurysm rupture is related to an affected function of the autophagy lysosomal system. Autophagy implies “self-eating” and can be understood as a process of lysosomal degradation of no longer usable cellular components. Therefore, it sets an important element of cellular metabolism [44].

Among the most serious consequences of intracranial aneurysm rupture are EBI, delayed brain injury and CVS, all associated with an impaired function of the autophagy-lysosomal system [44]. Aneurysm walls are usually characterized by an active inflammatory response, and NF-κB has been identified as the main transcription factor regulating the induction of inflammation-related genes in intracranial aneurysm lesions [45]. Moreover, NF-κB, which is a pivotal factor controlling inflammation, is regulated by autophagy-related proteins, and autophagy is regulated by NF-κB signaling. NF-κB denotes a family of transcription factors playing an important role in many cellular activities including proliferation, response to stress, immune response, apoptosis and inflammation [46]. Under stress conditions, NF-κB translocates to the nucleus, where it stimulates or represses the transcription of many genes. The activation of NF-κB in basal (normal) conditions can be promoted by two separate signaling pathways, the canonical and non-canonical pathway. The basal pathway can be stimulated by various factors, but in general the canonical pathway is activated by TNFα and IL-1β, as well as other cytokines via binding to their specific receptors [47]. The non-canonical pathway gets activated by a limited set of ligands, including those belonging to the TNF family and TNF-related factors. NF-κB, in an elevated activated form, generates the expression of several pro-inflammatory proteins, including cyclooxygenase-2 (COX-2), prostaglandin E2 (PGE-2) and molecules that facilitate the recruitment and adhesion of macrophages [45,48]. Once they have entered the vessel wall through new vasa vasorum, macrophages release other pro-inflammatory molecules, including TNFα, IL-1β, MMPs as well as other proteases, which finally results in the remodeling of the structure of the vessel wall in combination with other molecules [45,47,48]. An antiapoptotic effect was demonstrated due to the activation of autophagy with the immunosuppressive agent rapamycin (also named sirolimus), which is known as an inducer of autophagy undergoing several clinical trials in vascular diseases. In experimental aSAH, the activation of autophagy leads to decreased degree of CVS [49]. In contrast, an autophagy inhibition resulted in the deterioration of neurological deficits [50]. Therefore, autophagy and the NF-κB-mediated pathway might be other potential targets for neuroprotective therapies following aSAH, with special regard to the balance between physiologic and pathologic mechanisms of autophagy.

### 2.9. Meningeal Lymphatics

Caused by aSAH, blood, and accordingly red blood cell degradation products, rapidly enter the subarachnoid space and therefore the paravascular pathway, resulting in distinctly perivascular neuroinflammation [51]. A brain-wide pathway connecting perivascular spaces with CSF is the recently rediscovered system of lymphatic vessels, known as the glymphatic system, which is the utmost important in the clearance of metabolic waste products [52]. In addition to clots, lysis and phagocytosis by macrophages and neutrophils, meningeal lymphatic vessels drain CSF macromolecules and immune cells to cervical lymph nodes [53,54].

As mentioned earlier, after aSAH microglia is activated by the upregulation of inflammatory cytokines including TNFα, IL-1ß and IL-6, these are aggravated by the ablation of meningeal lymphatics, which leads to exacerbated neuroinflammation and neurological deficits. Therefore, the lymphatic system provides another promising opportunity in the development of therapeutic strategies regarding the treatment of brain injury after aSAH [55].

### 2.10. Novel Therapies Targeting Neuroinflammation

Inflammatory events that happen in the subarachnoid space can be divided into (1) cellular inflammation and (2) molecular inflammation. In addition, inflammatory processes take place in the three compartments (1) subarachnoid space, (2) vascular lumen and (3) cerebral parenchyma [3]. On the one hand, cellular components leave the blood vessels and enter the subarachnoid space. On the other hand, cellular and molecular factors act on the vascular walls. These findings raise the question of whether inflammation in the CSF is an outside–in or an inside–out reaction [3].

Considering an inflammatory response due to aSAH, a neuroprotective effect of early treatment with anti-inflammatory drugs as corticosteroids showed a diminished brain damage and a reduced mortality in a rat model [56]. Even a non-significant trend to improved neurological recovery could be demonstrated. However, several clinical studies could not confirm a beneficial effect of treatment with corticosteroids after aSAH in regard to CVS [57,58].

Another promising agent is human albumin, as it is known for anti-inflammatory properties [59,60]. The ALISAH study demonstrated neuroprotective effects due to a lower incidence of CVS and a trend of better neurological outcome [61].

A neuroprotective effect and reduced inflammation were recently demonstrated by the application of an IL-1 receptor antagonist (IL-1Ra) [62,63]. IL-6 was reduced in plasma and CSF, as IL-1Ra was able to cross the BBB. Therefore, IL-1Ra could be a safe neuroprotective agent.

Regarding useful potential biomarkers in the treatment of SAH patients as well as for prediction of outcome and post-SAH infections, serum IL-10 might be an additional useful parameter. SAH patients, who developed any kind of infection, CVS or chronic hydrocephalus, had significantly higher serum IL-10 levels compared to controls. In addition, discharged SAH patients with poor clinical outcome (modified ranking scale 3–6 or Glasgow outcome scale 1–3) revealed significantly higher serum IL-10 levels [64]. Another promising predictive biomarker correlating with clinical neurological outcome might be chemokine C-C motif ligand 5 (CCL5). Systemic and CSF CCL5 levels in aSAH patients on day 1 and day 7 could be independently associated with the clinical outcome at discharge. Therefore, CCL5 might also be another target for neuroprotective strategies in aSAH [65].

A potential neuroprotective agent regarding endocannabinoids might be the selective CB2R agonist JWH133. The application of JWH133 after experimental SAH achieved a protection of the BBB, shown in reduced brain edema and the attenuation of neurological outcome by suppressed leukocyte infiltration through TGF-1β upregulation and reduced neuronal apoptosis [66,67]. The same agent (JWH133) was also able to attenuate acute neurogenic pulmonary edema by preventing neutrophil migration after experimental SAH in rats [68]. In addition, the production of TNFα has been suppressed by cannabinoids leading to reduced CVS and brain ischemia after SAH [32]. In conflict with these results, a retrospective analysis showed a poor clinical outcome in SAH patients with verified cannabis consumption by the increased onset of delayed cerebral ischemia [69].

The challenge with regard to modulating inflammation is the fact that inflammation is often observed to be biphasic in nature, with elements that are both protective as well as deleterious. Identifying this temporal relationship and when to target involved pathways for therapeutic benefit remains a substantial challenge. Advanced neuroimaging may offer a viable option to detect biphasic peaks in the neuroinflammatory cascade. Finally, utilizing our current knowledge of SAH pathophysiology offers clear advantages therapeutically.

Table 1 summarizes new potential neuroprotective therapies in aSAH targeting neuroinflammation performed over the last 5 years. Despite the immense increase in the number of experimental, translational clinical data, some are still missing. A complementary collection of clinical neuroprotective therapies targeting neuroinflammation was nicely summarized by de Oliveira and Macdonald in 2018 [70]. Studies on neuroprotective therapies targeting neuroinflammation in aSAH older than 5 years have been collected and described by Lucke-Wold et al. in 2016 [14].

## 3. Role of Thromboinflammation in aSAH

### 3.1. Microthrombus Formation

Within the first few days after aSAH, delayed ischemia in cerebral microcirculation contributes to early brain injury and mortality [109,110,111,112]. Evidence for an early platelet activation after aSAH has been reported in both experimental and clinical aSAH [112,113]. Intravital microscopy of cerebral microcirculation could directly demonstrate the formation of platelet microthrombi in a mouse model of aSAH and showed that thrombus formation in cerebral microcirculation contributes to blood flow deficits [114,115]. Increased endothelial expression of P-selectin and platelet deposits could also be documented by immunohistochemical and electron microscopic studies [116]. Moreover, early microclot formation in cerebral microcirculation has been reported in a rabbit model of aSAH [117]. In addition, microemboli in small cerebral arteries in patients dying within 2 days after aSAH have also been documented in autopsies [118].

### 3.2. The von Willebrand Factor (vWF) and ADAMTS-13

Increased activity of vWF is associated with cerebrovascular thrombosis [119]. Concomitantly, the activity of the vWF-cleaving protease ADAMTS-13 (a disintegrin and metalloproteinase with a thrombospondin type 1 motif, member 13) decreases [119]. ADAMTS-13 has been described as a key protein in linking thrombosis with inflammation in injured brain [119,120,121]. Recombinant human ADAMTS-13 reduced microvascular thrombus formation and brain injury, when administered minutes after aSAH in both wildtype and ADAMTS−/− mice [122,123,124]. Interestingly, neuroinflammation, as monitored by the numbers of IBA-1 positive microglia in brain tissue, was reduced by recombinant human ADAMTS-13 and in vWF-deficient mice, whereas it was increased in ADAMTS-13 gene-deficient mice [119,123]. Plasma levels of ADAMTS-13 were reduced in the early phase after aSAH in patients with delayed cerebral ischemia [125]. In stroke patients treated with intravenous thrombolysis, low ADAMTS-13 plasma levels were associated with a worse outcome and high levels of inflammatory cytokines [126]. Thus, ADAMTS-13 has been identified as a potential biomarker for delayed cerebral ischemia after aSAH [127,128].

### 3.3. The Contact-Kinin System and Activated FXII

The contact-kinin pathway has been shown to play a role in the pathology of ischemic stroke and traumatic brain injury, not only by fostering vascular permeability and inflammation via kinins such as bradykinin, but also by promoting thrombus formation through the activation of the intrinsic pathway (also known as the contact pathway) [129,130]. Activated FXII (FXIIa) triggers the intrinsic coagulation cascade via the activation of FXI and induces a cascade of events by the cleavage of plasma kallikrein leading to the release of the inflammatory mediator bradykinin [129,130]. In experimental models of cerebral ischemia and traumatic brain injury, the genetic deletion of FXII or a pharmacological inhibition of FXIIa prevents microvascular thrombosis and neuroinflammation leading to reduced neuronal cell loss and improved functional neurological recovery [131,132,133,134,135,136]. Interestingly, the deficiency or pharmacologic blockade of FXII was reported to reduce neuroinflammation and render mice less susceptible to experimental autoimmune encephalomyelitis [137]. Remarkably, these beneficial effects are not accompanied by an increased risk of bleeding in experimental stroke or traumatic brain injury [131,132,135,138]. FXII deficiency actually reduced bleeding induced by tissue plasminogen activator (tPA) in experimental stroke [139]. Even if the effects of FXIIa inhibition in experimental models of aSAH have not been studied yet, therapies targeting FXIIa may also have translational potential in clinical aSAH.

### 3.4. Platelet Receptors

Apart from the FXIIa-mediated platelet activation, other platelet proteins contribute to inflammation after brain injuries [121,140]. Here, the platelet glycoprotein (GP)Ib-mediated platelet activation seems to be crucial for the interaction with immune cells. In experimental models of stroke, the activation of T-cell subsets contributes to brain damage in part by a platelet-dependent mechanism [141,142,143]. GPIb as a part of the GPIb-IX-V complex mediates the initial binding of platelets to the subendothelial matrix as a first step of thrombus formation. Inhibition of GPIb by neutralizing Fab fragments has anti-thrombotic and anti-inflammatory effects in mouse models of stroke [141,142,143]. Again, the beneficial effect was not associated with an increased risk of bleeding. Recent studies in mouse models of stroke have provided further evidence for T-cell interactions with platelets [144].

### 3.5. Potential Neuroprotective Approaches

The selective inhibition of the aforementioned pathophysiological events has been tested in experimental models of stroke, traumatic brain injury and aSAH as well as treatments targeting platelets [145,146,147,148,149], clot clearance [150], brain inflammation and immune cell infiltration [111,151,152,153,154]. These potential neuroprotective approaches are summarized in Table 2.

## 4. Role of Metabolism in aSAH

Similarly to the previously mentioned pathophysiological mechanisms and therapy targets, cerebral metabolism, which is already known for important pathological changes in various brain injuries such as TBI or stroke, came to the fore when a closer look was taken at aSAH pathophysiology [155]. Energy dysfunction arises in the early phase of aSAH and remains for a prolonged period of time after the event of bleeding. Therefore, research regarding the changes in post-aSAH energy metabolism inside the brain could help to understand the underlying pathophysiology of cerebral energy dysregulation in aSAH, investigating its impact on outcome and improve therapy management in clinical practice.

### 4.1. Metabolism with Regard to Early Brain Injury

A sudden bleeding in the subarachnoid space, commonly caused by an aneurysmal rupture, leads to a rapid increase in intracranial pressure (ICP), instantly followed by a reduction in cerebral perfusion pressure (CPP) and cerebral blood flow (CBF) [156]. While CPP recovers fast by a compensatory raise of mean artery blood pressure (MABP), CBF further decreases and stays below its normal levels for a prolonged period of time [8]. In addition to the decrease in CPP other mechanisms like ongoing acute artery vasoconstriction seem to cause cerebral hypoperfusion—a so-called low-flow-status of the brain. This sudden cerebral perfusion deficit ends up in multiple metabolic disturbances, one of a multitude of factors, which can lead to ischemic brain injury [156]. To maintain their normal function, neurons strictly depend on continuous supply with oxygen and glucose, the latter being the main source of energy for the brain. Via oxidative phosphorylation within the mitochondria, adequate amounts of adenosine triphosphate (ATP) were generated. In the case of acute cerebral ischemia, oxygen supply is disrupted and oxidative phosphorylation has to switch to insufficient anaerobic glycolysis in neurons and other brain-resident cells, leading to the accumulation of lactate in the form of an acidosis [157]. In combination with the reduced generation of ATP, this results in ion channel dysfunction, the disruption of normal cell membrane potential, production of reactive oxygen species and finally brain tissue damage by neuronal cell apoptosis [158].

Both preclinical and clinical trials repeatedly showed an increase in lactate, glutamate and lactate to the pyruvate ratio (LPR) in the interstitial brain tissue after aSAH, measured via cerebral microdialysis (CMD), as an indicator of brain metabolism derangement [8,159,160,161]. Nevertheless, the cellular level of metabolism disruption in neurons is still unknown and remains the subject of current investigations. Carpenter et al. found a significant reduction in global cerebral metabolic rate for oxygen (CMRO_2_) in 11 patients within the first few days after aSAH, measured via positron emission tomography (PET), independent of vasospasm-induced ischemia, hydrocephalus or intracerebral hematoma [162]. Further examinations of the course of hemodynamic parameters, CBF and tissue oxygenation (ptiO_2_) in an aSAH-animal model by Westermaier et al. detected an excess of tissue oxygenation several hours after aSAH, with any knowledge of prolonged increase in lactate, glutamate and LPR as metabolites of anaerobic metabolism pathways suggesting a disturbed cellular oxygen utilization and cerebral metabolic depression [163]. The basic idea is, that the affected cells are not able to use oxygen by oxidative glucose reduction via the metabolites pyruvate and acetyl-CoA, followed by the citric acid cycle and finally the respiratory chain. One or more steps of this energy metabolism pathway seem to be disturbed. In this context, Lilla et al. demonstrated for the first time a reduction in pyruvate dehydrogenase (PDH) activity following aSAH, independent of the supply of substrates [9]. As PDH is the key enzyme of the citric acid cycle, its activity reduction could be an independent factor contributing to a derangement of oxidative metabolism, a failure of oxygen utilization and secondary brain damage.

Furthermore, it is known that ischemic brain injury leads to the depolarization of the mitochondrial membrane potential, also causing a decrease in ATP production and the apoptosis of neuronal cells in the end [164]. Interestingly, Hayakawa et al. showed a release of functional mitochondria from astrocytes in the extracellular space after an experimental stroke in mice, finally entering neurons and triggering cell surviving signals [165]. Conversely, derangements in this mitochondria release pathway led to worse neurological outcome. According to that, further studies proved extracellular mitochondria in CSF both in rats and humans after aSAH, showing a significantly decreased mitochondrial membrane potential in the early after the aSAH phase [166]. Similar results were found in aSAH-patients with DCI [167]. Within the aSAH cohort, patients with higher mitochondrial membrane potential were even correlated with better neurological recovery 3 months after ictus [166]. In summary, these studies suggest that the extracellular mitochondria are a potential biomarker for the occurrence and recovery of brain injury. Due to lacking further studies, it remains unclear whether these endogenous neuroprotective mitochondrial transfer mechanisms may be exogenous therapeutical targets.

### 4.2. Metabolism with Regard to Delayed Brain Injury

One of the best studied mechanisms of delayed brain injury after aSAH is DCI, which commonly occurs 3–14 days after the bleeding event and represents a main cause for poor functional outcome after aSAH besides EBI. Until recently, the major cause of DCI has been thought to be CVS [168]. As vasospasm-targeted therapies in many clinical trials did not improve post-aSAH outcome, and that there were some indications that vasospasm and DCI occur separately in different locations, studies successfully looked for other DCI-underlying mechanisms [169,170]. As already mentioned, these mechanisms include, among others, cerebral vascular dysfunction, microthrombosis and neuroinflammation [171].

With this intention, experimental and clinical trials were performed to investigate the interdependency between cerebral energy metabolism and DCI. In comparison with the early phase after aSAH, similar results were received by using CMD: elevation of lactate, LPR and glutamate, while glucose and pyruvate levels were reduced in the post-aSAH long-term phase [172]. Interestingly, several studies detected significantly increased LPR and decreased glucose levels 12–16 h before DCI onset in aSAH patients, proposing these metabolites as indicators for DCI and potential help for preventing severe complications in the delayed phase of aSAH [173,174,175]. However, further studies are needed to investigate the possible treatment effects on energy metabolism and DCI as convincing results have barely existed until now. The limitations of the mentioned studies above, all using CMD as diagnostic tool, are restricted sensitivity due to the local measurement (ischemia distant to the monitoring catheter may not be detected) and restricted specificity because of the impossible differentiation between ischemic and non-ischemic metabolism derangements [172].

### 4.3. Potential Neuroprotective Approaches Targeting Metabolism

In the search for novel neuroprotective therapy options in aSAH, due to the illustrated mechanisms it seems that deranged cerebral metabolism could be a potential therapy target. Ca^2+^-channel blockers, like nimodipine, which are common part of therapy management in aSAH all over the world, were found to provide an advantage in terms of metabolic disruption, histological damage and clinical outcome after cerebral ischemia [176]. While the underlying mechanism is not cleared up yet, it may be worth having a closer look at the cellular level of metabolic disturbances. As Lilla et al. suggested, the inactivation of PDH could play a critical role in EBI after aSAH, so that preventing this inactivity may act as a neuroprotective factor [9]. Dichloroacetate (DCA) is a small molecule that crosses the BBB and stimulates PDH activity by the inhibition of PDH kinase [177]. Over the years, trials with different animal models inducing stroke or traumatic brain injury showed the neuroprotective potential of DCA, which became apparent in limiting lactate acidosis, the restoration of ATP, and improving the neurological outcome after hypoperfusion [178,179]. Kho et al. even found evidence that oxidative stress, the activation of microglia, BBB disturbance and even neuronal cell death in hypoglycemia-induced ischemia are decreased by DCA [180]. As the synopsis of all these findings, one may draw the conclusion that DCA supports metabolic recovery and therefore raises the ischemic limit of brain cells at risk of neuronal death. As DCA has not been investigated in aSAH models to date, a transfer to this disease as novel neuroprotective therapy option could be revealing.

As it is capable of the restoration of the oxidative cell metabolism, a neuroprotective effect of acetyl-L-carnitine (ALCAR) was also proven in multiple both experimental and clinical studies [181,182,183,184]. The acetylcarnitine-CoA transferase helps ALCAR entering the citric acid cycle, corresponding to a “bypass” of the PDH reaction. Therefore, the application of ALCAR could significantly lower the brain lactate level, restore ATP and even improve the neurological outcome [182,185].

In addition, mitochondrial dysfunction after aSAH is shown to activate the autophagy of neuronal cells, again one of a multitude of factors leading to EBI and DCI [44]. Therefore, targeting the autophagy–lysosomal system could be a conceivable therapy approach. This system seems to prevent cell surviving mechanisms, as long as its function is appropriate. From the moment that lysosomal-triggered autophagy gets out of control, i.e., after aSAH, an increase in the rate of cell death can be detected [186]. On this occasion worth mentioning, Chen et al. investigated the effect of epigallocatechin-3-gallate (EGCG) by an experimental study, coming to the conclusion that this active metabolite of tea catechin has the potential to lessen mitochondrial membrane potential depolarization, autophagy dysfunction and ultimately even neurological deficits [187]. Again, concerning this matter, further studies are urgently needed.

## 5. Role of Cerebral Vasospasm in aSAH

### 5.1. Pathophysiology of Cerebral Vasospasm

The exact pathophysiology of CVS occurrence is still not completely understood, while many different suggested mechanisms such as prolonged smooth muscle contraction, endothelial damage or increased endothlin-1 production exist [188].

Nevertheless, CVS demonstrates the response to damaged blood vessels due to the degradation products of red blood cells and secondary inflammation-induced vasoconstriction as the activation of the neutrophils and production of cytokines, as mentioned earlier. Moreover, different phases of CVS, including an early and a delayed response, also seem to exist [28].

In contrast to macrovascular factors, furthermore, changes in microcirculation especially seem to play a significant role in the development of delayed cerebral ischemia [189].

Endothelin 1 is known for its vasoconstrictive properties and the fact that it is overproduced in aSAH. Thus, it initially appeared to be a promising target for the treatment of CVS. Subsequently, several trials were initiated to investigate the effect of endothelin receptor antagonists. The probably most popular agent is clazosentan [190]. The phase II CONSCIOUS-1 trial demonstrated a dose-dependent reduction in CVS under medication with clazosentan [191]. The following phase III trial (CONSCIOUS-2) failed to show an improvement of the clinical outcome while the downstreamed CONSCIOUS-3 trial had to be stopped previously due to concerning side effects under therapy with clazosentan [192,193]. In summary, despite a role in reducing CVS, no clinical benefit was verified under medication with endothelin receptor antagonists.

### 5.2. Potential Neuroprotective Approaches Targeting Cerebral Vasospasm and Hypoperfusion

#### 5.2.1. Magnesium Sulfate

Due to a loss of ATP and ischemic depolarization, a cellular calcium influx results after aSAH. Therefore, calcium antagonists became an interesting target in the prevention of CVS.

A deficit of serum magnesium results in increased secondary cerebral ischemia. Magnesium ameliorates rheological function and dilates blood vessels as it works as a natural calcium blocker. Hypomagnesemia in aSAH patients was found to be correlated with the amount of blood in the subarachnoid space and neurological status. When developing during medical treatment, hypomagnesemia correlates with ischemic infarctions [194].

Oral intake does not increase the serum concentrations significantly.

Several clinical studies examined the effects of magnesium treatment after aSAH with partially conflicting results [195]. One of the reservations refers to insufficiently crossing the blood–brain barrier. While several studies demonstrated a reduction in delayed cerebral ischemia and an ameliorated neurological outcome [196,197,198,199], two large trials failed to show a reduction in secondary ischemia as well as the improvement of clinical outcome [200,201], most likely dependent on the co-medication with nimodipine which works as a competitive antagonist.

Even though just one trial could show neuroprotective results, treatment with nimodipine [202], a dihydropyridine calcium antagonist, is still established worldwide.

Recent results of two clinical trials with the intracisternal application of magnesium sulfate demonstrated significantly less CVS or delayed cerebral ischemia while establishing a better functional outcome in contrast to the control groups. The additional administration of intravenous hydrogen could further show supplementary effects by finding reduced liquor levels of neuron-specific enolase, as a marker of neuronal injury, as well as reduced malondialdehyde, which is an indicator of oxidative stress [203,204].

#### 5.2.2. Hypercapnia

After aSAH, the period of expected CVS has its maximum between day 4 and 14 due to suspended autoregulation. Under physiological conditions, the self-regulating mechanism is able to adapt to changes of arterial blood pressure for keeping the CBF constant, inter alia, by arterial partial pressure of carbon dioxide (P_a_CO_2_). Subsequently, a therapeutic use of changes in P_a_CO_2_ to increase the CBF was investigated in the clinical studies of our group [205,206]. It could be shown that within a clinical study including 12 patients with poor grad aSAH, CBF reproducibly increased during controlled phases of hypercapnia and remained raised within the first hour after downgrading to baseline without generating rebound effects resulting in a low incidence of secondary infarctions and a relatively good neurological outcome. The side effect of a mild increase in the intracranial pressure due to enhanced CBF was buffered by CSF drainage and failed to reach a pathological level according to the investigation by Petridis et al. studying the effect of permissive hypercapnia in aSAH patients [207].

The promising results of this non-pharmacological treatment will be further evaluated in a randomized multicenter trial.

In a dose optimization study of our group, temporary hypercapnia of 45 min was verified to be the optimum duration for therapeutic use (unpublished own data, manuscript submitted).

#### 5.2.3. Hypothermia

A neuroprotective effect due to mild hypothermia was shown in a few experimental trials [208,209,210]. ICP was significantly lowered and CBF ameliorated. Even a reduced rate of injured neurons was shown, though it remains unclear whether hypothermia causes an attenuating effect or only delays brain injury. In patients with severe aSAH, therapeutic hypothermia achieved a reduction in arterial flow velocity [211]. Neuroprotective targets, potential agents and therapeutic strategies in different compartments following aSAH are summarized in Figure 1.

## 6. Conclusions

Aneurysmal subarachnoid hemorrhage continues to be a difficult complex cerebrovascular disease with a consistent limitation of pharmacological treatment options. Morbidity and mortality remain high despite the implementation of promising therapies such as nimodipine for treating cerebral vasospasm, new mechanisms of pathophysiology of aSAH occurred-like inflammatory processes or metabolic derangements. Having a closer look at these mechanisms, it is clear that several dysregulations take place in different compartments—vessel wall, subarachnoid space, brain parenchyma and cellular level—at different points in time—EBI/DBI, early and delayed vasospasm. Thus, several brain injury pathways must be influenced at the right place and preferably the right time to optimize therapeutic efficacy in general. While therapeutical strategies at the metabolic level are only in their early phase, no standard of care could be established yet with anti-inflammatory strategies. The non-pharmacological opportunities are promising, especially targeting vasospasm and reducing DCI for a better functional outcome. Translational clinical data should notably be in focus of future research. As beyond our scope, this review does not point out every pathophysiological aspect that is or has been under investigation in aSAH research.

Since no therapeutical breakthrough in aSAH has been made to date, and as expected further research is needed, it is vital to develop an idea of its consequences in terms of its outcome and developing potential therapies efficiently targeting brain injury.

## Figures and Tables

**Figure 1 ijms-22-05442-f001:**
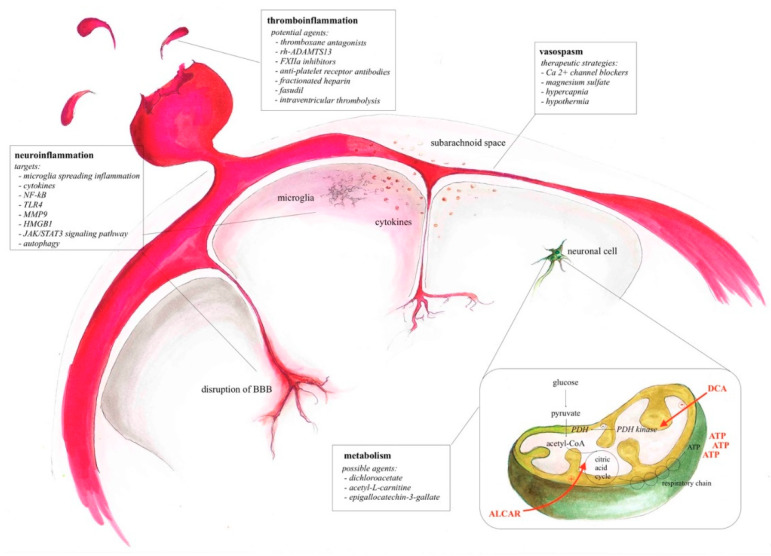
Summary of neuroprotective targets, potential agents and therapeutic strategies in different compartments following aSAH.

**Table 1 ijms-22-05442-t001:** Potential neuroprotective therapies targeting neuroinflammation.

Therapeutic Agent	Target	Model	Outcome Measures/Findings	Reference
LPS, PLX3397	Microglia	Experimental aSAH in mice	Reduction in neuronal cell death	Heinz, R. et al., 2021 [29]
Interleukin-2 (IL-2)	Regulatory T-cells	Experimental aSAH in rats	Reduction in neuronal injury and proinflammatory factors, increase in neuronal functions	Dong et al., 2021 [71]
Mesenchymal stem cell-derived extracellular vesicles	Microglial M2 polarization	Experimental aSAH in rats	Reduction in inflammatory cytokines and inflammation, increase in neuroprotective effects	Han et al., 2021 [72]
Adenosine A3 receptor agonist	Microglial polarization	Experimental aSAH in rats	Increase in anti-inflammatory and neuroprotective effects	Li et al., 2020 [73]
Milk fat globule-epidural growth factor (MFG-EP)	Microglial polarization	Experimental aSAH in mice	Reduction in brain edema and proinflammatory factors, increase in neurological factors	Gao et al., 2021 [74]
Mixture of gas containing argon	Microglial inflammatory response	Experimental aSAH in rats	Reduction in early hippocampal neuronal damage	Kremer et al., 2020 [75]
EPZ6438 (specific EZH2 inhibitor)	EZH2 (enhancer of zeste homolog 2)	Experimental aSAH in rats	Reduction in attenuated neuroinflammatory brain injury	Luo et al., 2020 [76]
Oxyhemoglobin (OxyHb)	RNF26 (regulating TLRs)	Experimental aSAH in rats	Silence: reduction in neuronal injury and neurological dysfunction	Chen et al., 2020 [77]
LP-17	TREM-1 myeloid cells	Experimental aSAH in mice	Amelioration of microglial pyroptosis	Xu et al., 2020 [78]
Resolvin D1	Lipoxin A4 receptor/formyl peptide receptor 2 (ALX/FPR2) in microglia	Experimental aSAH in rats	Inhibiting H6 promoted microglial pro-inflammatory polarization, neuronal oxidant damage and death	Liu et al., 2020 [79]
Hydrogen sulfide (H_2_S)	TLR4/NF-κB pathway in microglia	Experimental aSAH in rats	Reduction in cognitive impairment and amelioration of neuroinflammation in microglia	Duan et al., 2020 [80]
Oxyhemoglobin (OxyHb)	CC chemokine ligand 20 (CCL20)	Experimental aSAH in mice	Reduction in apoptotic neurons	Liao et al., 2020 [81]
Translocator protein (TSPO) and TSPO ligand Ro5–4864	Microglia/macrophages polarization	Experimental aSAH in mice	Improvement of neurological function, increase in expression of anti-inflammatory factors	Zhou et al., 2020 [82]
Oxyhemoglobin (OxyHb)	CCM3 overexpression and NF-κB signaling pathway	Experimental aSAH in rats	Reduction in cellular degeneration, neurocognitive impairment and inflammatory factors (TNF-a and IL-1β)	Peng et al., 2020 [83]
Modified exosomes (miR-193b-3p)	HDAC3, NF-κB	1. aSAH patients and healthy controls to define profile 2. experimental aSAH in mice	Reduction in homological behavioral impairment, brain edema and BBB injury	Lai et al., 2020 [84]
Curcumin	M2 polarization through TLR4/MyD88/NF-κB signaling pathway	Experimental aSAH in tlr4^−/−^ mice and wild type (WT)	Alleviation of neuroinflammation response, microglia phenotype shift and release of proinflammatory mediators	Gao et al., 2019 [85]
Dehydroepiandrosterone (DHEA)	Microglial activation	Experimental aSAH in C57BL/6 mice	Increase in neuroprotective effects, suppression of inflammation	Tao et al., 2019 [86]
Apelin-13	Apelin receptor (APJ)/endoplasmic reticulum stress associated inflammation	Experimental aSAH in rats	Reduction in oxidative stress and neuroinflammation	Xu et al., 2019 [87]
BMS-470539	Melanocortin 1 receptor (MC1R)	Experimental aSAH in rats	Suppression of microglial activation and neutrophil infiltration	Xu et al., 2020 [88]
TAK 242 (TLR4 antagonist)	Toll-like 4 receptor (TLR4)	Experimental aSAH in mice	Suppression of brain edema	Okada et al., 2020 [89]
Bexarotene	Retinoid X receptor	Experimental aSAH in rats	Decrease in neuroinflammation, improvement of neurological deficits	Zuo et al., 2019 [90]
RP 001 hydrochloride	S1P/S1PR pathway	Experimental aSAH in mice	Decrease in neuroinflammation, alleviation of neurological damage	Li et al., 2019 [91]
Bone marrow mesenchymal stem cells	Notch 1 signaling pathway	Experimental aSAH in rats	Amelioration of neurobehavioral impairments and BBB disruption	Liu et al., 2019 [92]
Fluoxetine	TLR4/MYD88/NF-κB pathway	Experimental aSAH in rats	Decrease in BBB disruption and brain edema, improvement of neurological function	Liu et al., 2018 [93]
Apolipoprotein E	Jak2/STAT3 signaling pathway	Experimental aSAH in mice	Decrease in oxidative stress and inflammation	Pang et al., 2018 [94]
TSG-6	Microglial phenotype shift/SOCS3/STAT3 pathway	Experimental aSAH in rats	Amelioration of brain injury, decrease in proinflammatory mediators	Li et al., 2018 [95]
TAT-Pep5P	Resident microglia, p75 neurotrophin receptor (p75NTR)	Experimental aSAH in transgenic mice	Reduction in microglial activation, neuroinflammation and EBI	Xu et al., 2019 [96]
MST1 inhibitor XMU-MP-1	MST1, NF-κB/MMP-9 pathway	Experimental aSAH in mice	Alleviation of neurological deficits, BBB, brain edema, neuroinflammation and white matter injury	Qu et al., 2018 [97]
rh-Aggf1	PI3K/Akt/NF-κB pathway	Experimental aSAH in rats	Decrease in neuroinflammation and BBB disruption, improvement of neurological deficits	Zhu et al., 2018 [98]
IAXO-102 (TLR4 antagonist)	TLR4	Experimental aSAH in C57BL/6 mice	Reduction in neurological impairments, brain edema, BBB disruption, increase in survival rates	Okada et al., 2019 [99]
Bexarotene	PPARγ	Experimental aSAH in C57BL/6 mice	Increase in neurological function, reduction in neuronal cell death and microglial activation	Tu et al., 2018 [100]
FTY720 (PP2A agonist)	Tristetraprolin (TTP), protein phosphatase 2A (PP2A)	Experimental aSAH in rats	Reduction in apoptosis, neuroinflammation and brain edema, increase in neurological function	Yin et al., 2018 [101]
Rolipram (specific phosphodiesterase-4 inhibitor)	SIRT1/NF-κB pathway	Experimental aSAH in rats	Reduction in brain edema, neurological dysfunction and neuronal cell death	Peng et al., 2018 [102]
Human Netrin-1 (rh-NTN-1)	UNC5B (receptor of NTN-1)	Experimental aSAH in rats	Increase in neurobehavioral function, reduction in brain edema and microglia activation	Xie et al., 2018 [103]
Fluoxetine, AC-YVAD-CMK (caspase-inhibitor)	NLRP3 inflammasome, caspase-1	Experimental aSAH in rats	Increase in neurological function, reduction in brain edema and autophagy activation	Li et al., 2017 [104]
Methylene blue	Akt/GSK-3β/MEF2D pathway	Experimental aSAH in rats	Reduction in neurological dysfunction and brain edema	Xu et al., 2017 [105]
IL-1 receptor antagonist (IL-1Ra, anakinra)	Interleukin-1 (IL-1)	Randomized, open-label, clinical study in aSAH-patients	Difference in plasma IL-6, plasma pharmacokinetics for IL-1Ra, clinical outcome at 6 months	Galea et al., 2018 [63]
AE1–329 (EP4 selective agonist)	Prostanoid 4 receptor (EP4)	Experimental aSAH in rats	Reduction in neurological dysfunction, BBB damage, brain edema, reactivation of microglia, proinflammatory cytokines	Xu et al., 2017 [106]
Rutin	RAGE- NF-κB inflammatory signaling pathway	Experimental aSAH in rats	Increase in neurological function, reduction in BBB permeability, brain water content and neuronal cell death	Hao et al., 2016 [107]
Exogenous LXA4 (lipoxin A4)	Formyl peptide receptor 2 (FPR2), p38 MAPK	Experimental aSAH in rats	Increase in neurological functions, reduction in neutrophil infiltration and brain water content	Guo et al., 2016 [108]

**Table 2 ijms-22-05442-t002:** Potential neuroprotective pharmacotherapies targeting thromboinflammation.

Therapeutic Agent	Target	Model	Reference
Thromboxane antagonists, COX1-inhibitors, PAF antagonists	Platelet aggregation	Experimental and clinical aSAH	Lagier et al. [149], Suzuki et al. [145], Tokiyoshi et al. [146], Hirashima et al. [147,148]
intraventricular thrombolysis (rh tPA)	Clot clearance	Experimental and clinical aSAH	Shi et al. [150]
rh-ADAMTS13	vWF-induced thrombosis and inflammation	Experimental and clinical aSAH	Muroi et al. [122], Vergouwen et al. [123,125], Wan et al. [124], Chauhan et al. [120]
FXIIa inhibitors (C1 inhibitor, rh infestin-4)	Contact kinin system (platelet aggregation and neuroinflammation)	Experimental stroke and TBI	Kleinschnitz et al. [131], Hagedorn et al. [132], Heydenreich et al. [133], Hopp et al. [135,136], Albert-Weissenberger et al. [134]
Anti-platelet receptor antibodies	Thrombosis, neuroinflammation, immune cells	Experimental stroke and TBI	Kleinschnitz et al. [141], Schuhmann et al. [143], Albert-Weissenberger et al. [140], Stoll and Nieswandt [121]
Fractionated heparin, glibenclamide, statins, anti-proinflammatory cytokine agents	Neuroinflammation	Experimental and clinical aSAH	James et al. [153], McBride et al. [111], Vergouwen et al. [151]
Fasudil (ROCK2 inhibitor)	Neuroinflammation	Experimental intracerebral hemorrhage (ICH) and clinical aSAH	McBride et al. [111], Li et al. [154], Zhao et al. [152]
Nimodipine	Vasospasms, thrombosis, leukocyte infiltration	Experimental and clinical aSAH	McBride et al. [111]

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
