# Peer review of "Neuroprotective Strategies in Aneurysmal Subarachnoid Hemorrhage (aSAH)"

_ijms, 2021, doi:10.3390/ijms22115442_

Round 1
Reviewer 1 Report
The review article from Weiland et al. summarized the data on neuroprotective therapies in SAH. The manuscript is well written. The language is clear and understandable. There are some typing errors. As a whole, it has a good balance between being comprehensive and detailed.
The article properly addresses the raised questions. The different approaches to the neuroprotective strategies have been well explained. It nicely covers the current status of therapeutic options from different aspects.
Though, the addition of some recent papers on chemokine mediated Inflammation in SAH (Chaudhry et al 2020, Cytokine) and Cytomkine IL-10 in nosocomial infections after SAH (Chaudhry et al 2020, IJMS) will be valuable.
Author Response
Tha valuable recent papers on chemokine mediated inflammation in SAH (Chaudhry et al. 2020, Cytokine) and IL-10 in nosocomial infections after SAH (Chaudhry et al. 2020, IJMS) have been added and integrated in the manusctipt on page 6, line 21-31.
Reviewer 2 Report
The review should be updated with new neuroprotective strategies, such as the use of endocannabinoids.
Author Response
The neuroprotective potential of endocannabinoids in neurodegenetaive and chronic inflammatory diseases (like parkinson's disease, MS, ALS and epilepsy) is surely a hot scientific topic these days.
Regarding SAH, only a few papers can be found between the years 2014-2016.
However, we added another chapter "2.5 endocannabinoids" which can be found on page 3, line 36 up to page 4, line 10 as well as a section on chapter 2.10 novel therapies targeting neuroinflammation which can be found on page 6, line 32-41.
Round 2
Reviewer 2 Report
The article is suitable for publication.